# Adaptation and Validation of the Multidimensional Fairness Scale in Chilean University Students

**DOI:** 10.3390/bs14121148

**Published:** 2024-11-29

**Authors:** Fredy Cea-Leiva, Sergio Dominguez-Lara, Sonia Salvo-Garrido

**Affiliations:** 1Program de Doctorado en Ciencias Sociales, Universidad de La Frontera, Temuco 4811230, Chile; fcea@uct.cl; 2Escuela de Posgrado, Universidad Católica de Temuco, Temuco 4781509, Chile; 3Instituto de Investigación FCCTP, Universidad de San Martín de Porres, Lima 15047, Peru; sdominguezl@usmp.pe; 4Departamento de Matemática y Estadística, Universidad de La Frontera, Temuco 4811230, Chile

**Keywords:** fairness, distributive justice, procedural justice, emerging adulthood, measurement

## Abstract

The Multidimensional Fairness Scale (MFS) assesses an individual’s experience of fairness across the many contexts of daily life. It has been applied in the USA; however, the psychometric qualities of reliability and validity have not been examined in a Spanish-speaking population or among Chilean university students. A cross-sectional study was conducted on 377 university students to explore these properties. Students from public (46.2%) and private (53.8%) universities participated, with a mean age = 21.5 (SD = 3.3). CFA was performed, showing good goodness-of-fit indices (RMSEA = 0.08; CFI = 0.97; TLI = 0.96) for the model of three related factors (interpersonal, occupational, and societal) and good reliability indices. Although changes were made to the original structure, the tested model was consistent with the theoretical structure of the MFS, which allows its use on the Chilean university population.

## 1. Introduction

Justice studies have a long trajectory within a multi-disciplinary context [1]; therefore, we find a long and sometimes contradictory use of concepts. While there are various definitions of social justice [2,3,4,5], here, it is understood as the “promotion of a fair and equitable allocation of bargaining powers, resources, and obligations in society in consideration of differential power, needs, and abilities of people to express their desires” [6] (p. 754). In short, fairness is understood as the application of distributive, procedural [7], and interactional justice.

Justice seems to be a moral imperative based on the modern idea that we are all born free and equal and have the same rights and obligations [8]. The notion of justice goes back to the Judeo-Christian biblical idea that God grants the good or evil deserved in a person’s life [1]. Authors associate Aristotle with the relationship between justice and inequality, mentioning that an unequal distribution due to differences in merit is just [9]. Hobbes, Locke, and Rousseau contend that the state or society formulates regulations that delineate the forms of equality aimed at fostering integration and social order [9]. Rawl [10] proposes a normative framework to create a fair structure in society, where the main social institutions regulate the distribution and social burdens, emphasizing those less favored.

Justice is subjective [9]; it is a social construct and, as such, may (or may not) be influenced by objective events (such as inequality in income or treatment), as well as by affective subjective elements [11].

Empirical studies point to two perspectives on a subject’s evaluation of justice. The first is the utopian perspective, which shows that assessments of what is fair depend on an individual’s normative orientation [12,13] or an ideal of fairness [14], also known as order attitudes [8]. These are rules or norms that people think should guide the allocation and (re)distribution of goods and burdens within social groups. Thus, there are four attitudes or principles: Equality (everyone is allocated the same); equity (benefits and burdens are allocated according to individual contributions and efforts); need (benefits are allocated taking care of those most in need) [15]; and entitlement (benefits and burdens are allocated maintaining the status quo as they consider, for example, social origin, race, or sex) [12].

The second is the existential perspective, where judgments invoke existential norms [13]. Here, it is suggested that there are individual differences in distributive preferences influenced by the comparison with their immediate context [16], which is analyzed by income justice theories, such as relative deprivation [17] and equity theory [18]. In this regard, Berger [19] points out, for example, that an individual believes he should have the same as others like him.

Although wealth inequality has increased [20], it can be considered a threat to the stability of governments [21], since it would cause greater social pressure for redistribution, situations of instability, and crises of legitimacy [22]. This argument is questioned by some studies since, to produce instability and redistributive pressures, inequalities must be perceived [14]. This perception is influenced by biases [16,23], but these levels of inequality must also be deemed unjust [16]. Integration, stability, and even the absence of unrest in a society depend on a concept of justice, which, if accepted by its members, gives legitimacy to social institutions and the state [24]. Merit is one of the mechanisms of legitimization of inequality, especially in societies where an individualistic ideology of society predominates [25], and redistributive policies would be a mechanism of legitimization in more collaborative societies.

There are various forms of justice [26], and despite some distinctions, they can generally be categorized into three main types: distributive, procedural, and interactional justice. Distributive justice is the most studied [16]; it refers to allocating benefits and burdens across social groups [9]. Procedural justice focuses on the decision-making processes that shape such allocations [27,28]. Interactional justice focuses on how people deal with each other on a daily basis [29].

Concerning distributive justice, Ng and Allen [30] mention four theories on how individuals make distributional judgments in the economy in general: The self-interest theory, where judgments depend on how much they benefit personally; the belief in a just world theory, where judgments are affected by the need to believe in an orderly, stable, and just world; the attribution theory, where judgments depend on whether the outcome is attributed to an internal or external cause; and finally, the ideological perspective, where judgments are coerced by ideologies that prescribe values, attitudes, and behaviors outside the individual.

With respect to interactional justice, it is perhaps the most perceived and most rejected by the population. It can be considered a barometer of how other inequalities are interpreted and how judgments about society are constructed [31]. This refers to the dignified and respectful treatment between people [9,29] and impacts adherence to the collective. In Chile, Araujo [31], in his studies of the everyday in public transportation, speaks of a malignancy of otherness, of seeing the other as an enemy, which affects the construction of the public and collective.

Social science studies have shown that experiences of injustice affect attitudes and behavior and produce social consequences that can affect the functioning of organizations and institutions [9], voting intentions, redistributive preferences, social unrest or migration [32], political trust [33], and the legitimacy of social institutions [24].

The perception of inequality correlates positively with higher income and higher education levels [14], contrary to other studies that indicate that the perception is higher among those with lower income [34]. Evidence also links a heightened perception of inequality with present and future views of stagnant or worsening economic performance [35], as well as with those who identify with left-wing parties [36,37].

Concerning evaluations of what is fair, in the normatively oriented framework, higher-income groups prefer equity, and lower-income groups prefer equality [38]. Less educated groups prefer the principles of equality and need but also favor the principle of equity [12]. At the political level, conservatives prefer equity and tend to perceive poverty and unemployment as fair, whereas liberals prefer equality and perceive poverty and unemployment as unfair [39]. Furthermore, welfare state supporters believe that the economy distributes resources unfairly. In contrast, free-market supporters believe that the economy distributes equitably [30] or that equality is the guiding principle in more collectivist societies and equity in more individualistic societies [9]. In contrast, Norton and Ariely [40] point out that perceptions of fairness are not associated with public policy preferences.

Studies on procedural justice indicate that it is associated with greater public cooperation, respect for legal compliance [41], and greater support for governance [28].

Distributive and procedural justice relate to personal, interpersonal, organizational, and community well-being [42]. Experiences of injustice have been associated with mental health problems, depression, and drug use [43]. At the organizational level, unfair treatment is associated with lower job satisfaction [44], health problems, and burnout [45]. Hicks [46] asserts that fairness is a fundamental component of dignity.

At the gender level, studies report that women tend to support the idea of equality more, while men support the principle of equity [47]. It is also noted that women with traditional gender ideologies perceive less injustice than those with non-traditional gender ideologies, since the latter tend to share their husbands’ assessment that inequalities are unfair [48]. Segovia [49] points out that men, married people, and part-time workers perceive greater inequality.

In October 2019, a major social movement took place in Chile. More than 1.2 million people participated in a march in the capital of Chile and acts of vandalism such as looting of supermarkets and the burning of Santiago metro stations, which put the institutionality and the government of the time at risk [50]. The unrest observed can be attributed to the injustices and calls for greater dignity voiced by citizens [51]. Thus, valid and reliable instruments to assess fairness are essential, which is the aim of this study.

Instruments for measuring fairness vary according to the theoretical concept being assessed. Most instruments used to measure fairness are weak in their psychometric properties, and it is quite common to measure complex concepts using a single item [52].

An important stream is research on belief in a just world (BJW), seen by some as a system-justifying ideology, as it normalizes injustices by rationalizing the status quo and reducing anxiety, guilt, and uncertainty [53]. Here, we use Rubin and Peplau’s Belief in a Just World Scale [54], which is unidimensional, but various studies show from one to five dimensions [53]. Based on this development, Lipkus [55] created the Global Belief in a Just World (GBJWS), which measures belief in a just world through one factor and is probably one of the most widely used scales today.

Studies related to the perception of economic inequality commonly use three instruments. The first is through a Likert-type response item of the type “income differences are too great”. The second type asks about the income between occupations with higher and lower social prestige. The third shows images with their respective descriptions and asks which represents the country’s distribution [49]. Torres-Harding et al. [56] developed the Social Justice Scale (SJS) to measure favorable attitudes toward intentions to engage in social action using four dimensions: attitudes toward social justice, perceptions of behavior toward social justice, subjective norms of social justice, and behavioral intentions. Wegener and Liebig [57] proposed an instrument to measure justice attitudes through 12 items and four dimensions based on the four justice ideologies: equity, equality, need, and entitlement. Zhang and Zhou [28] proposed the Social Justice Scale using two items: one to measure procedural justice and the other to measure distributive justice. The Bertelsmann Stiftung has operationalized social justice via the Social Justice Index (SJI), which delineates six domains of justice and assesses 30 quantitative and 8 qualitative indicators, encompassing poverty prevention, equitable education, labor market access, social cohesion, non-discrimination, and health [58].

There are several developments in measuring distributive, procedural, and interactional justice in education, particularly with regard to fair assessment. For more details on instruments, see the systematic review by Rasooli et al. [59], which highlights the instrument of Abdelzadeh et al. [60], which measures procedural justice with three items and has good goodness-of-fit indicators. In addition, the instrument by Trip et al. [61] measures distributive and procedural justice with 21 items.

This research uses Duff’s Multidimensional Fairness Scale (MFS) [52], which includes distributive, procedural, and interactional justice items. The original version has 12 items and four dimensions, using a bifactor model, with general equity as a General Factor (GF) and four life domains as Specific Factors (SF): interpersonal, occupational, community, and societal. The objective is to measure an individual’s experience of fairness in the various and diverse manifestations of their daily life. Despite the fit indicators from the original study [52], which provides validity evidence in support of the bifactor structure (RMSEA = 0.07; CFI = 0.99; TLI = 0.99), it is important to note that the authors did not evaluate an oblique model to support the use of a higher-order model, since it is understood that this model is based on a simpler one. It is also likely that the fit indices have only been included to indicate that it is the best structure, which could be questioned because this type of analysis (with higher order factors) typically yields better fit indices than oblique models. Therefore, it is necessary to complement it with other indicators.

Notably, if a strong correlation exists among the SFs, it may be posited that a GF accounts for the variation in the items independently of that attributed to the SF [62]. Bifactor modeling is pertinent in these instances [63,64] as it aims to simultaneously evaluate the impact of the GF and the SFs on the items to ascertain the strength of the GF. This is achieved through the use of a variety of indicators, including the ECV (explained common variance) [65], which denotes the extent to which the common variance is accounted for by the presence of the GF; the hierarchical omega of the GF (ωh) [66], which reports the total variance attributed to the GF; and the hierarchical omega by dimensions (ωhs), which is interpreted as the variance of the factors even in the presence of the GF [64]. Then, the GF is significant if the ωh is greater than 0.70 and ECV is greater than 0.60 [64]. Conversely, the SFs are significant if the ωhs is greater than 0.30 and the ECV is below 0.50 [67,68]. This would thus provide information about a scale’s structure, whether it is unidimensional or multidimensional [69].

In this vein, a complementary analysis was performed to calculate the above indicators using the factor loadings provided in Table 8 of the original study [52]. Determining that the GF is not sufficiently supported (LCS = 0.572) is feasible. At the same time, the interpersonal (ωhs = 0.450) and occupational (ωhs = 0.589) dimensions can be interpreted independently of the GF, unlike the community (ωhs = 0.190) and societal (ωhs = 0.194) dimensions.

Despite the methodological limitations observed, the instrument is a useful tool to measure the current context in Chile, with high expectations for greater social justice, more horizontal treatment, and demands for better redistribution of resources [50]. It collaborates in two critical areas: on the one hand, it proposes an instrument with more than one item per dimension [52] and with psychometric evidence, which is frequently a deficiency in the instruments used in Chile, particularly in the social sciences [70]. On the other hand, this instrument collaborates in theoretical development by looking at justice from a perspective that incorporates distributive, procedural, and interactional justice and relates them to key elements of everyday life: interpersonal, occupational, community, and societal. This could contribute to generating a complex view of social justice in Chile, helping to explain the negative evaluation of the current state of society [71]. In addition, there are no other psychometric studies besides the original, which increases the need for a Spanish version adapted in Chile.

The general objective is to provide psychometric evidence for the MFS. The specific objectives are: (i) to evaluate the validity evidence based on the internal structure of the scale and (ii) to assess its reliability. Based on these objectives, two hypotheses are proposed: (H1) the MFS has a multidimensional structure, and (H2) the reliability of the MFS among Chilean university students is adequate.

## 2. Materials and Methods

### 2.1. Participants

The population of this study included university students from Temuco (N = 53,346). The whole population was not contacted. Non-probabilistic convenience sampling was conducted until an adequate sample size was reached. The minimum sample size (N = 200) was determined based on specific guidelines, taking into account the effect size, with 0.50 set as the minimum value for factor loadings [72], since that magnitude ensures that at least 25% of the item’s variance is explained by the factor. The desired statistical power level (1 − β = 0.80), probability level (α = 0.05), number of factors (4), and items (12) were implemented [73].

Responses were invited via email and in person. A total of 377 students volunteered and answered all questions. The ages of the participants were between 18 and 50 (M = 21.5 years; SD = 3.3). The sample consisted of 240 participants (63.7%) who identified themselves as female, 130 (34.5%) as male, and 7 (1.8%) as another gender. Moreover, 174 students (46.2%) were enrolled in public universities, while 203 (53.8%) were enrolled in private universities, studying in fields including social sciences and humanities (16.7%), engineering (14.1%), health (31.6%), and education (37.7%).

### 2.2. Instruments

The Multidimensional Fairness Scale (MFS) constructed by Duff [52] measures fairness at the experiential level in multiple life domains through twelve 5-point ordinal response items with the following options: 1 = never, 2 = rarely, 3 = sometimes, 4 = frequently, and 5 = always. Its original version presented a bifactor structure of general equity as a general factor with 4 life domains as specific factors: interpersonal, occupational, community, and societal. Each of the factors is composed of 3 items. An example question, which assesses community fairness, reads: When it comes to your experiences in your local community, how often do you feel that you have the same amount of privileges as everyone else? The MFS was validated in the American population and exhibited favorable psychometric properties [52]; however, Chile has not been subjected to any validity assessments. Furthermore, a sociodemographic questionnaire was employed, comprising questions about age, gender, and other factors.

### 2.3. Procedure

The scale was adapted and validated following the guidelines set by the International Test Commission [74]. After the scale was reviewed, the author was contacted for authorization to use it. It was subsequently translated and back-translated to adapt it into Spanish, with the assistance of experts and native speakers to assess any potential linguistic or cultural differences and the comprehension of the items. In the item, “you receive the same amount that you put in”, the experts recommended incorporating the words “time” and “monetary resources” in parentheses. In the header, “When it comes to your main occupation”, they recommended including “or studies”. A pilot test was also carried out with 15 students from various faculties to assess their understanding of the instructions and the items, a process that culminated in no new changes.

For data collection, the students were reached via email and in person through a QR code, which directed them to Question Pro (advanced version), a digital platform where the instruments were hosted. Informed consent forms were given to all participants to safeguard the ethical principles of the project. This study was approved by the Science Ethics Committee of the Universidad de La Frontera (File No.126_22; Research Protocol Page No.094/22).

### 2.4. Data Analysis

The analysis was conducted in stages, taking into account the specific objectives. To fulfill the first specific objective, the most appropriate factor structure of the MFS was established by evaluating theoretically grounded models using confirmatory factor analyses (CFA). Figure 1 shows the theoretical models to be tested with the data: the one-factor model (model 1), four-factor oblique model (model 2), and three-factor oblique model (model 2b).

Multivariate outliers were initially identified with the Mahalanobis distance (*p* < 0.001) [75]. Univariate normality was assessed according to skewness (<2) and kurtosis (<7) [76], and multivariate normality was assessed with the G2 coefficient (<70) [77].

The assessment was performed with the weighted least square mean and variance adjusted (WLSMV) estimation method [78], which is recommended for the use of ordinal variables [79] over a wide range of sample sizes [80]. Moreover, WLSMV assumes no distributional assumptions about the observed variables [81]. Consequently, the robust standard errors of the structural coefficients are more accurate than those obtained with MLR and ULSMV in all asymmetric data situations [82]. The goodness-of-fit was evaluated using the following indicators: comparative fit index (CFI), Tucker–Lewis index (TLI), root mean square error of approximation (RMSEA), and the standardized root mean square residual (SRMR). This last indicator was used as a complement to the RMSEA since new research has shown that SRMR outperforms RMSEA when the data evaluated are of a categorical nature [83]. An adequate fit is assumed when the CFI and TLI present values greater than 0.90 [84] to assess the models. Values lower than 0.08 are adequate for the RMSEA [85]. For the SRMR, a value lower than 0.08 is considered a good fit [86]. Similarly, the magnitude of the factor loadings was also considered, with magnitudes above 0.30, although factor loadings above 0.50 are preferred [72], as the item would have higher quality to reflect the construct because it would explain an acceptable amount of item variance (25%). In this sense, with higher factor loadings, the construct would be better represented in the responses of that item.

Finally, reliability was assessed using Cronbach’s alpha coefficient (α) [87] and McDonald’s omega (ω) [88].The Mplus 7.11 program [89] was used.

## 3. Results

### 3.1. Descriptive Analysis

Table 1 presents the descriptive analysis (mean, standard deviation, skewness, and kurtosis) conducted on the 12 items and factors of the scale. The magnitude of skewness and kurtosis was adequate and multivariate normality had favorable evidence (G2 = 18.764).

### 3.2. Factor Structure

The goodness-of-fit indices for the unidimensional structure (model 1) were inadequate (CFI = 0.765, TLI = 0.713, RMSEA = 0.206, CI 90% 0.195–0.218, SRMR = 0.106). In contrast, the four-factor structure (model 2) substantially improved the fit (CFI = 0.965, TLI = 0.952, RMSEA = 0.085, CI 90% 0.072–0.098, SRMR = 0.044), but the correlation between factor 3 (community) and factor 4 (societal) was high (0.922), so the decision was made to test a variation (model 2a) including the items corresponding to these factors in a single factor (CFI = 0.966, TLI = 0.956, RMSEA = 0.080, CI 90% 0.068–0.094, SRMR = 0.045). Likewise, the factor loadings were acceptable (Table 2). Finally, adequate evidence of reliability was noted in every case (Table 2).

## 4. Discussion

The aim of this study was to assess the factor structure and reliability of the Multidimensional Fairness Scale (MFS) in a sample of Chilean university students. The original bifactor structure that included general equity as the GF with four SFs (interpersonal, occupational, community, and societal) [7,52] was questioned in this study since, according to the complementary analysis performed with the information from the original manuscript, the GF does not have sufficient support to scale to a higher-order model. Therefore, a four-factor correlated model and not a bifactor was proposed for the original study. The most appropriate model for the current study was a three-factor correlated model, which maintained the 12 items of the original scale, but consolidated the community and societal domains into a single factor. In addition, interfactor correlations were moderate, which does not support modeling a GF under a bifactor approach.

It is important to emphasize the 12 items on the scale that were maintained, as they exhibited loads that exceeded the limit value. This reinforces the fundamental theory that examines distributive justice and procedural justice from an ecological perspective that spans the interpersonal, occupational, community, and societal domains. This multifaceted approach allows for a more comprehensive and accurate assessment of how individuals perceive justice in various aspects of their lives. This differs from the original study, in which the existence of a GF was proposed under a correct theoretical approach, but without sufficient empirical support.

Likewise, the original study only mentioned distributive and procedural justice [52], but additional studies have separated treatment justice from procedural justice [9]. In the Chilean context, this is especially relevant given that one of the main citizen demands raised in the 2019 social movement was greater dignity [51] and higher expectations for horizontal treatment in everyday life [31]. However, despite not being declared an interactional justice dimension, the MFS addresses this aspect with questions such as: “You are treated with dignity and respect”.

The major change in the structure of the MFS is the joining of the community and societal dimensions, which were separate in the original version. Chilean society took a neoliberal turn in 1973 with the establishment of the civil–military dictatorship of Augusto Pinochet. Since then, the community has been characterized by fragmentation, containment, bureaucratization, and depoliticization at the organizational [90] and spatial levels. In this context, items related to education and health are perceived as general topics and are, therefore, grouped coherently in the social dimension. In the 2019 mobilizations, the right to health and education were among the main demands connected to those for justice [51]. This finding illustrates the extent to which the historical and social context can influence the perceptions of justice in particular contexts.

Regarding the conceptual contributions of this study, the subjective component [9] of social justice is reaffirmed and may (or may not) be influenced by objective elements like income inequality, which could be explored in subsequent studies. Despite the fact that inequality is regarded as a factor that undermines democracies [21], the reinterpretation of other injustices [31] at the level of procedures and results is influenced by experiences of injustice of all types, particularly those that occur in ordinary life. This research contributes to the literature by emphasizing the significance of interactional justice and its influence on the overall perception of justice.

The MFS contributes two theoretical elements: on the one hand, looking at and defining social justice as a complex construct determined by distributive, procedural, and interactional justice, and on the other, considering that these three types of justice concern objective and subjective resources [26], which highlights the importance of looking at experiences of justice in different facets of life [52], such as interpersonal, occupational, and society in general. This comprehensive approach is crucial to developing policies and practices that address the multiple dimensions of social justice.

In the social sciences, there is a challenge in measuring latent constructs; often, complex constructs are measured with a single item, and psychometric properties of reliability and validity are not observed in the scales used [70]. This research provides evidence of the validity and reliability of the MFS, offering an instrument that addresses the perception of social fairness, including distributive, procedural, and interactional justice within a multifaceted and complex approach that is especially relevant to the Chilean context, which experienced a social outbreak where one of the most pressing demands was for greater justice and dignity in treatment [51].

A primary limitation of the study concerns the representativeness of the sample. Participation was limited to students from universities in Temuco, primarily from the Araucanía region (southern Chile), which accounts for approximately 6% of the national population. Future research should involve probability samples that represent different age groups and cities across the country. Furthermore, future research should consider incorporating a variety of universities and socioeconomic contexts to guarantee that the findings are applicable to a broader population. It would also be beneficial to investigate the potential impact of other variables, such as political orientation and ethnic identity, on perceptions of fairness in order to gain a deeper and more nuanced understanding of this construct in various sociocultural contexts.

## Figures and Tables

**Figure 1 behavsci-14-01148-f001:**
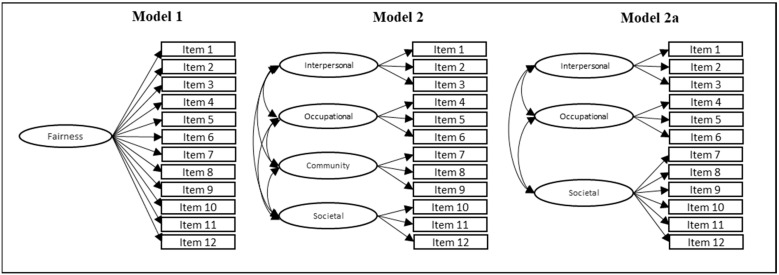
Models for the MFS.

**Table 1 behavsci-14-01148-t001:** Descriptive statistics.

		M	SD	g1	g2
Item 1	You receive the same amount that you put in (Entregas y recibes de manera recíproca (tiempo, recursos monetarios, sentimientos))	3.72	1.01	−0.44	−0.58
Item 2	You are treated with dignity and respect (Eres tratado con dignidad y respeto)	4.16	0.84	−1.03	1.04
Item 3	You are listened to (Eres escuchado)	3.78	0.97	−0.55	−0.23
Item 4	You are fairly rewarded for your effort at your workplace (Eres justamente recompensado por tu esfuerzo)	3.19	0.95	−0.22	−0.40
Item 5	You are consulted on important matters (Se te consulta sobre asuntos importantes)	3.14	1.07	−0.30	−0.57
Item 6	You participate in decision making (Participas en la toma de decisions)	3.30	1.09	−0.29	−0.68
Item 7	You have opportunities to obtain a good education (Tienes oportunidades de obtener una buena educación)	3.89	0.89	−0.54	−0.15
Item 8	You have the same amount of privileges as everyone else (Tienes la misma cantidad de privilegios que los demás)	2.99	1.11	−0.06	−0.67
Item 9	You are able to access good healthcare (Puedes tener acceso a una buena atención médica)	3.18	1.12	−0.03	−0.81
Item 10	You receive your fair share in society (Recibes tu parte justa en la sociedad)	2.96	0.89	0.06	−0.06
Item 11	You receive the same opportunities as others in society (Recibes las mismas oportunidades que los demás en la Sociedad)	2.93	1.02	0.16	−0.53
Item 12	Your voice counts in society (Tu voz es escuchada en la Sociedad)	2.55	1.02	0.15	−0.63
Factor 1	Interpersonal	3.89	0.79	−0.57	0.03
Factor 2	Occupational	3.21	0.85	−0.16	−0.50
Factor 3	Societal	3.08	0.74	0.04	−0.31

Notes. Means (M), standard deviations (SD), skewness (g1), kurtosis (g2), ( ) items in Spanish.

**Table 2 behavsci-14-01148-t002:** Standardized factor loadings. CFA model with three specific factors.

Items	F1	F2	F3
Item 1	0.662		
Item 2	0.890		
Item 3	0.858		
Item 4		0.702	
Item 5		0.841	
Item 6		0.731	
Item 7			0.578
Item 8			0.661
Item 9			0.599
Item 10			0.826
Item 11			0.824
Item 12			0.771
F1	1		
F2	0.572	1	
F3	0.435	0.567	1
Reliability			
ω	0.849	0.803	0.862
α	0.792	0.751	0.823

Note: interpersonal (F1), occupational (F2), societal (F3).

## Data Availability

Data are contained within the article.

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
