# Peer review of "Adaptation and Validation of the Multidimensional Fairness Scale in Chilean University Students"

_behavsci, 2024, doi:10.3390/bs14121148_

Round 1

Reviewer 1 Report

Comments and Suggestions for Authors

Just a few suggestions for improvement

1) Please explain the response scales against 12 items. Was it categorical or numerical? You have used number for analysis. Please explain maximum and minimum numerical range in the response. 

2) It would have been useful if you could provide some description of the sample characteristics. You have used volunteering samples from university population.  It would be useful if you can limit your claims only on this population. 

Reviewer 2 Report

Comments and Suggestions for Authors

Thank you for submitting this well-written paper. The article offers a valuable contribution by focusing on the adaptation, localization, and validation of a scale originally developed and utilized in the U.S., now applied to Chilean university students. This research holds significant importance for the field of psychometrics. Some suggestions are as follows: 1) In the paragraph, "In October 2019, a major social movement took place in Chile that put the institutionality and the government of the time at risk," I recommend providing more context and details about the event. Expanding on this will help establish a stronger rationale for the need for valid and reliable instruments to measure fairness, which is the focus of your study. 2) In the paragraph, "Given the current context in Chile, the instrument is a useful tool despite...," I suggest elaborating on the specific context in Chile. It would be beneficial to explain why this instrument is particularly relevant for the Chilean setting and what factors led to the decision to adapt it for Chilean university students. This additional detail will enhance the reader's understanding of the instrument's significance in relation to the local context. 3) I am interested in the consent procedure you implemented for the participants, as well as the total number of students who took part in the study. Out of the 53,346 students, only 377 agreed to participate? Could you provide more context regarding this participation rate? Additionally, could you clarify the method you used to determine that a sample size of n=200 is the minimum required? Please specify, as this information is important for understanding the study's validity. 4) You mentioned that you conducted a pilot study after the scale was adapted and translated. I suggest you provide details on how the pilot study was carried out, including the number of students involved. Were these participants also Chilean university students? Sharing this information will enhance the clarity and rigor of your research methodology. 5) I recommend expanding the discussion section to elaborate on the scholarly significance of this study. Overall, I believe this paper is well-written, and the statistics section appears to be solid.

Comments on the Quality of English Language

The overall quality of the English language used in this paper is strong. 
